# NiH-catalyzed asymmetric hydroarylation of *N*-acyl enamines to chiral benzylamines

Yuli He [1], Huayue Song[1], Jian Chen[1] & Shaolin Zhu [1✉]

Enantiomerically pure chiral amines and related amide derivatives are privilege motifs in many pharmacologically active molecules. In comparison to the well-established hydro-amination, the transition metal-catalysed asymmetric hydrofunctionalization of enamines provides a complementary approach for their construction. Here we report a NiH-catalysed enantio- and regioselective reductive hydroarylation of *N*-acyl enamines, allowing for the practical access to a broad range of structurally diverse, enantioenriched benzylamines under mild, operationally simple reaction conditions.

---

[1] State Key Laboratory of Coordination Chemistry, Jiangsu Key Laboratory of Advanced Organic Materials, Chemistry and Biomedicine Innovation Center (ChemBIC), School of Chemistry and Chemical Engineering, Nanjing University, Nanjing 210093, China. ✉email: shaolinzhu@nju.edu.cn

As a privileged structural motif, benzylamines and related amide derivatives are found in many natural products, pharmaceuticals, agrochemicals, and other chemicals (Fig. 1a), and efficient strategies for their catalytic, enantioselective synthesis have long been sought[1–5]. Metal hydride[6–9] catalyzed reductive hydrofunctionalization from readily available alkene starting materials is a particularly appealing approach to the synthesis of benzylamines. Previously, starting from styrene and an electrophilic amination reagent, Buchwald[10] and Miura and Hirano[11] have independently developed an enantioselective reductive CuH-catalyzed hydroamination method (Fig. 1b, left). We recognized that if asymmetric hydroarylation of enamines could be achieved, enantioenriched benzylamines would become accessible (Fig. 1b, right).

Benefiting from the economical and facile chain-walking and cross-coupling[12–14], and use of simple ligands, NiH catalysis has emerged in recent years as an efficient means of achieving enantioselective C–C bond formation[15–58]. In these general synthetic processes: (1) both of the starting alkenes and aryl halides/alkyl halides are commercially or synthetically available; (2) no prior generation of organometallic reagents is necessary; and (3) the newly formed sp³-hybridized stereocenters could potentially be enantioselectively controlled at the carbons originating in the achiral olefins[39–43] or at the carbons from racemic alkyl electrophiles[35–38]. Recently, we reported the enantioselective hydroarylation of styrenes using a novel chiral nickel-bis(imidazoline) catalyst (Fig. 1c, i)[40]. In this process, an asymmetric center was generated and controlled at the carbon derived from the olefin. To demonstrate the wide-ranging applicability of this reductive NiH[54–58] catalysis, we have explored the feasibility of asymmetric hydroarylation with electron-rich alkenes, for example N-acyl enamines, which are generally less reactive than styrenes (Fig. 1c, ii).

As shown in Fig. 1c, ii, the syn-hydrometallation of an L*NiH species into an N-acyl enamine would generate two alkyl-nickel enantiomers. These could undergo oxidative addition with an aryl iodide, affording two high-valent Ar-Ni(III)-alkyl enantiomers, which would experience rapid homolysis and sequential stereoselective radical recombination prior to a selective reductive elimination[59–70]. In the presence of a suitable chiral ligand, the radical recombination process could be enantioselectively controlled and deliver a single Ar-Ni(III)-alkyl enantiomer in an enantioconvergent fashion. Subsequent reductive elimination would deliver the enantiopure arylation product[69]. Notably, the amide group in the enamine substrate would also play a key role, enhancing both the regio- and the enantioselectivity. Here we describe the successful execution of this strategy, which allows for the practical access to a broad scope of chiral benzylamines under mild, operationally simple reaction conditions. During the preparation of this report, similar work was reported by Nevado et al.[42,43].

## Results

**Reaction design and optimization**. Our initial studies focused on the enantioselective hydroarylation of enamide (1a) with 4-iodoanisole (2a), and obtained the results summarized in Fig. 2 (see Supplementary Tables 1, 2 for details). After extensive examination of nickel sources, ligands, silanes, bases, and solvents, we found that NiI₂ and chiral bis-imidazoline ligand (L1) can provide the desired hydroarylation product in good yield as a single regioisomer with high enantioselectivity (99% ee, entry 1). Other nickel sources such as NiBr₂ led to lower yields with almost no change in ee (entry 2). Evaluation of ligands showed that both the imidazoline skeleton (entry 3 vs entry 1) and the remote steric effects of the substituent on the imidazoline skeleton (entry 4 vs

entry 3) have a marked influence on the enantioselectivity[70]. Dimethoxymethylsilane was shown to be an unsuitable silane (entry 5) and KF was shown to be an unsuitable base (entry 6). Use of DMF as solvent also led to a significantly lower yield (entry 7) and use of the less-polar THF as solvent produced no desired arylation product (entry 8). Reducing the reaction time from 48 h to 24 h led simply to incomplete conversion (entry 9).

**Substrate scope**. Having established the optimal conditions, we explored the scope of the aryl iodide coupling partner (Fig. 3) and found that a wide range of aryl and heteroaryl iodides are tolerated. The aryl substituent can be substituted at the ortho, meta, or para position (2a–2p), however, aryl iodide with an ortho-

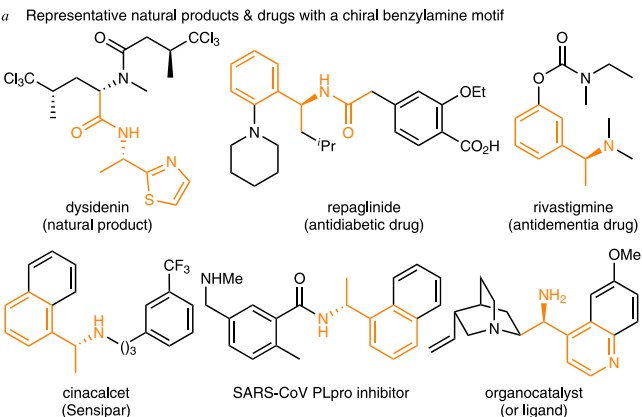

a  Representative natural products & drugs with a chiral benzylamine motif

dysidenin
(natural product)

repaglinide
(antidiabetic drug)

rivastigmine
(antidementia drug)

cinacalcet
(Sensipar)

SARS-CoV PLpro inhibitor

organocatalyst
(or ligand)

b  Two reductive hydrofunctionalization strategies for their synthesis

c  NiH-catalysed enantioselective reductive hydrofunctionalization of alkenes

i  Our previous work: NiH-catalysed enantioselective hydroarylation of styrenes

ii  This work: NiH-catalysed enantioselective hydroarylation of N-acyl enamines

**Fig. 1 Asymmetric hydroarylation of *N*-acyl enamines to access chiral benzylamines. a** Representative natural products & drugs with a chiral benzylamine motif. **b** Two reductive hydrofunctionalization strategies for their synthesis. **c** NiH-catalyzed enantioselective reductive hydrofunctionalization of alkenes.

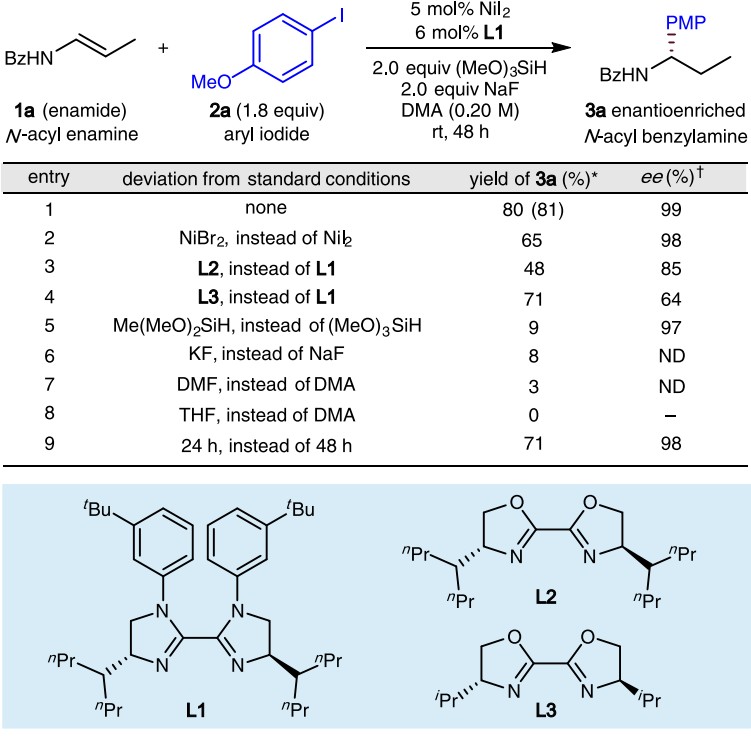

| entry | deviation from standard conditions | yield of **3a** (%)* | ee (%)[†] |
|-------|-----------------------------------|---------------------|-----------|
| 1 | none | 80 (81) | 99 |
| 2 | NiBr$_2$, instead of NiI$_2$ | 65 | 98 |
| 3 | **L2**, instead of **L1** | 48 | 85 |
| 4 | **L3**, instead of **L1** | 71 | 64 |
| 5 | Me(MeO)$_2$SiH, instead of (MeO)$_3$SiH | 9 | 97 |
| 6 | KF, instead of NaF | 8 | ND |
| 7 | DMF, instead of DMA | 3 | ND |
| 8 | THF, instead of DMA | 0 | — |
| 9 | 24 h, instead of 48 h | 71 | 98 |

**Fig. 2 Variation of reaction parameters.** *Yields determined by crude $^1$H NMR using 1,1,2,2-tetrachloroethane as the internal standard, the yield in parentheses is the isolated yield. [†]Enantioselectivity was determined by chiral HPLC analysis. Bz benzoyl, PMP p-methoxyphenyl, DMA N,N-dimethylacetamide, DMF N,N-dimethylformamide.

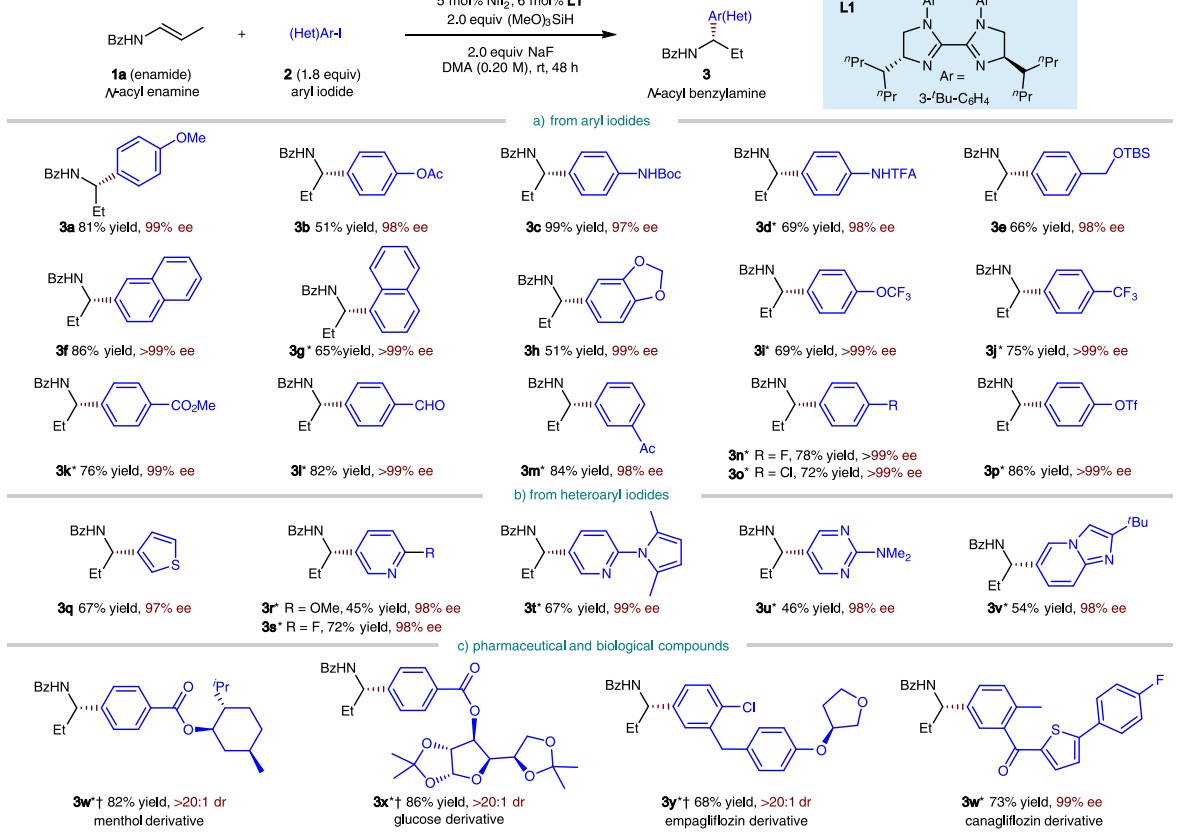

**Fig. 3 Substrate scope of aryl iodide component.** Yield under each product refers to the isolated yield of purified product (0.20 mmol scale, average of two runs), >95:5 regioisomeric ratio (rr) unless otherwise noted. Enantioselectivities were determined by chiral HPLC analysis. *5 mol% Ni(ClO$_4$)$_2$•6H$_2$O, DMA (0.10 M), 1.5 equiv ArI. [†]Diastereoisomeric ratio (dr) was determined by crude $^1$H NMR analysis. TBS, tert-butyldimethylsilyl.

**Fig. 4 Substrate scope of the *N*-acyl enamine component.** Yield and ee are as defined in Fig. 3 legend. *Ac* acetyl.

substituent is less reactive (**2g**, see Supplementary Figs. 1, 2 for the details of competition experiment). Substrates with Electron-rich (**2a**–**2e**) as well as electron-withdrawing (**2h**–**2p**, **2w**–**2z**) aryl iodides work well in the reaction. In case of the latter, Ni (ClO₄)₂·6H₂O was found to be a superior catalyst and only 1.5 equiv of aryl iodide was needed. A variety of functional groups, including ethers (**2a**, **2e**, **2h**, **2i**, and **2y**), esters (**2b**, **2k**, **2w**, and **2x**), a carbamate (**2c**), an amide (**2d**), a trifluoromethyl group (**2j**), aryl fluorides (**2n**, **2z**), as well as a ketal (**2x**), are all readily accommodated. Notably, sensitive functional groups such as an easily reduced aldehyde (**2l**) and ketone (**2m**, **2z**), a chloride (**2o**, **2y**), and a triflate (**2p**) commonly used for subsequent cross-coupling all remained unchanged under the exceptionally mild reaction conditions of the reaction. Compounds containing heterocycles such as thiophene (**2q**, **2z**), pyridine (**2r**, **2s**, and **2t**), pyrrole (**2t**), pyrimidine (**2u**), and imidazopyridine (**2v**) are also competent coupling partners. With this protocol, several core structures of bioactive and pharmaceutical molecules, such as L-menthol (**2w**), glucose (**2x**), empagliflozin (**2y**), and canagliflozin (**2z**), could be readily introduced in an enantioselective fashion, irrespective of the existing chiral centers and complex structures.

As shown in Fig. 4, the scope of the enamide is also fairly broad. In general, high levels of enantioselectivity are delivered by the reaction. For *N*-benzoyl enamine substates, an electron-deficient substituent on the aromatic ring of the benzoyl group led to a higher yield than electron-rich substituents (**1c** vs **1b**). The less sterically hindered *N*-acetyl enamine (**1e**) was more reactive than *N*-pivaloyl enamine (**1d**, see also Supplementary Fig. 3 for X-ray structure of **4d**). The β-unsubstituted enamide (**1d**) was also shown to be a viable substrate. Enamides with a range of different functionalized alkyl substituents at the

β-position underwent asymmetric hydroarylation smoothly (**1h**–**1m**). A diverse spectrum of functional groups were compatible, including ethers (**1i**, **1j**), esters (**1k**, **1l**), and an alkyl chloride (**1m**). In addition, both *E* and *Z* isomers of the enamide substrates produced the same enantiomeric product with the same level of enantioselectivity ((*E*)-**1h** vs (*Z*)-**1h**).

## Discussion

The robustness and synthetic utility of this catalytic system were further demonstrated by gram-scale synthesis and subsequent derivatization of the product (Fig. 5a). A 5 mmol-scale hydro-arylation was performed successfully and the product (**3a**) was readily converted into the tertiary amine (**5a**) without racemization. To shed light on the hydrometallation process, deuterium-labeling experiments were carried out with deuteropinacolborane (Fig. 5b). From both *E* and *Z* isomers of the enamide substrates, a diastereomeric mixture of deuterated products were obtained with an opposite dr ratio. If the *syn*-hydrometallation of NiD to *N*-acyl enamine is the enantio-determining step, then a diastereomerically pure **4h-D** should be formed. The observed formation of both diastereoisomers in each case indicates that the NiD insertion is not the enantio-determining step (As shown in Supplementary Figs. 4–9, we could observe the isomerization of *E* olefinic substrate to *Z* isomer during the reaction process. In contrast, the isomerization of *Z* olefinic substrate to *E* isomer is very slow.). On the other hand, the same level of enantioselectivity for deuterated products in both cases of *E* and *Z* olefinic substrates (Fig. 5b) is consistent with a mechanism in which rapid homolysis of Ni(III) to Ni(II) and the subsequent enantioselective radical recombination serves as an enantio-determining step (Fig. 1c, ii).

**Fig. 5 Gram-scale, derivatization, and deuterium-labeling experiments. a** Gram-scale experiment and reduction of the amide. **b** NiD experiment: NiD *syn*-hydrometallation is not the enantio-determining step.

In conclusion, we have developed an enantioselective hydro-arylation of *N*-acyl enamines, which provides access to an array of enantioenriched benzylamines, a biologically active pharmaco-phore. This reaction is based on a reductive NiH catalysis strat-egy. A wide range of functional groups on both the *N*-acyl enamine and aryl iodide components are well-tolerated. Pre-liminary studies of the mechanism suggest that the hydro-metallation of NiH is not the enantio-determining step. Development of a migratory version of this transformation and investigations of the mechanism are currently in progress.

## Methods

**General procedure for NiH-catalyzed asymmetric hydroarylation of *N*-acyl enamines**. In a nitrogen-filled glove box, to an oven-dried 8 mL screw-cap vial equipped with a magnetic stir bar was added $NiI_2$ (3.1 mg, 5.0 mol%), **L1** (7.2 mg, 6.0 mol%), NaF (16.8 mg, 2.0 equiv), and anhydrous DMA (1.0 mL). The mixture was stirred for 20 min at room temperature, at which time (*E*)-*N*-(prop-1-en-1-yl) benzamide (**1a**) (32.2 mg, 0.20 mmol, 1.0 equiv), 4-iodoanisole (84.0 mg, 0.36 mmol, 1.8 equiv), and $(MeO)_3SiH$ (51.0 μL, 0.40 mmol, 2.0 equiv) were added to the resulting mixture in this order. The tube was sealed with a teflon-lined screw cap, removed from the glove box and the reaction was stirred at rt (22~26 °C) for up to 48 h (the mixture was stirred at 750 rpm, ensuring that the base was uni-formly suspended). After the reaction was complete, the reaction mixture was directly filtered through a short pad of silica gel (using EtOAc in Petroleum ether) to give the crude product. 1,1,2,2-Tetrachloroethane (26 μL, 41 mg, 0.25 mmol) was added as internal standard for $^1$H NMR analysis of the crude material. The product was purified by chromatography on silica gel for each substrate. The yields reported are the average of at least two experiments, unless otherwise indicated. The enantiomeric excesses (% ee) were determined by high-performance liquid chro-matography analysis using chiral stationary phases.

## Data availability

The authors declare that the main data supporting the findings of this study, including experimental procedures and compound characterization, are available within the article and its supplementary information files, or from the corresponding author upon reasonable request. CCDC 2036489 contains the supplementary crystallographic data for **4d**. These data can be obtained free of charge from The Cambridge Crystallographic Data Centre via www.ccdc.cam.ac.uk/data_request/cif.

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

## Acknowledgements

Research reported in this publication was supported by NSFC (21822105, 21772087, and 22001118), NSF of Jiangsu Province (BK20200300, BK20201245), programs for high-level entrepreneurial and innovative talents introduction of Jiangsu Province (group program), and Fundamental Research Funds for the Central Universities (020514380182). The authors thank Zouhong Xu (professor Yi Lu and professor Weiyin Sun group) and Penglong Wang (professor Congqing Zhu group) for assistance with X-ray structure determination.

## Author contributions

Y.H. and S.Z. designed the project. Y.H., H.S., J.C., and S.Z. co-wrote the manuscript, analyzed the data, discussed the results, and commented on the manuscript. Y.H., H.S., and J.C. performed the experiments. All authors contributed to discussions.

## Competing interests

The authors declare no competing interests.
