## [Peer Review File · Nature Communications]

REVIEWER COMMENTS

Reviewer #1 (Remarks to the Author):

Zhu and coworkers described a Ni-H-catalyzed asymmetric hydroarylation reaction of N-acyl enamines for the preparation of optically active benzylamines. The reaction is well designed upon their previous achievements in NiH-catalyzed asymmetric hydroarylation reaction of styrenes. The carefully tuned chiral bis-imidazoline ligand enables this reaction to achieve high yield and enantioselectivity. In comparison with previous methods, the Ni-H-catalyzed reaction was performed under mild conditions and displayed high functional tolerance and structural diversity. Ethers, esters, ketal, ketone and aldehyde groups are tolerable. Furthermore, the geometry of enamides does not affect the enantioselectivity of the products. I would suggest this work be published in *Nat. Commun.* after considering the following minor points:

- (1) This reviewer agrees with the authors' statement that "the amide group in the enamine substrate would also play a key role, enhancing both the regio- and the enantioselectivity". Is there any evidence that the Ni(III) in Fig. 1c(II) shows significant radical character while not the Ni(I), generated after syn-addition? The NiD-mediated reaction indicates that the NiD insertion is not the enantio-determining step. Is it possible that the isomerization to the desired enantiomer takes place at the Ni(I) species?
- (2) Could the authors give some explanation for the role of NaF while KF shows poor reactivity?
- (3) Is there any chemoselectivity for C-I bonds with different steric environments, such as 1,4-diiodo-2-methylbenzene?
- (4) In the Supporting Information, the ¹³C NMR spectroscopies were not properly referenced. They are not inconsistent with the statement "are referenced CDCl₃ at 77.16 ppm".
- (5) Page S136, the compound 1k was contaminated by impurities.
- (6) There are shoulder peaks in 3e, page S152. Please double check!

Reviewer #2 (Remarks to the Author):

Zhu and co-workers have reported a method for the enantioselective synthesis of benzylic amine derivatives from vinyl enamines and aryl halides. The method is very simple and effective. Independently, a related work by the Nevado group just appeared in *ACIE* "doi.org/10.1002/anie.202011342". However, the reaction conditions used in this paper have much better reaction efficacy in terms of scope, functional group compatibility, yields, enantioselectivities as well as substrates (e.g. internal olefins are working well). The authors also show that the products can be obtained effectively independent of the E/Z isomers of the substrates. Given the operational simplicity, general applicability, valuable product formation, and the surge of this emerging NiH catalysis area, I strongly recommend this work in *Nature Communication* after a minor revision.

Specific comments:

1. What about other N-Protected functional groups (e.g. NHCbz, NHBoc)?
2. "Evaluation of ligands showed that both the imidazoline skeleton (entry 3 vs entry 1) and the remote steric effects of the substituent on the imidazoline skeleton (entry 4 vs entry 3) have a dramatic influence on the enantioselectivity." How does it affect the reactivity and enantioselectivity? Why is the bis-imidazoline ligand more effective than the bi-ox ligand? Some explanations are welcome.
3. What is the role of Na⁺ ion in this reaction?
4. In SI, the yields of the synthesis of the substrates are missing.
5. How did the author establish the syn- and anti-configuration of the 4h-D?
6. The authors have an intriguing observation. E- and Z isomers give similar EEs. Can the authors check if there are relevant isomerization processes?

In response to **reviewer 1** (quotes from reviewer are italicized):

Reviewer #1 (Remarks to the Author):

Zhu and coworkers described a Ni-H-catalyzed asymmetric hydroarylation reaction of N-acyl enamines for the preparation of optically active benzylamines. The reaction is well designed upon their previous achievements in NiH-catalyzed asymmetric hydroarylation reaction of styrenes. The careful tuned chiral bis-imidazoline ligand enable this reaction to achieve high yield and enantioselectivity. In comparison with previous methods, the Ni-H-catalyzed reaction was performed under mild conditions and displayed high functional tolerance and structural diversity. Ethers, esters, ketal, ketone and aldehyde groups are tolerable. Furthermore, the geometry of enamides does not affect the enantioselectivity of the products. I would suggest this work publish in Nat. Commun. after considering the following minor points:

1. This reviewer agrees the authors' statement that "the amide group in the enamine substrate would also play a key role, enhancing both the regio- and the enantioselectivity". Is there any evidence that the Ni(III) in Fig. 1c(II) shows significant radical character while not the Ni(I), generated after syn-addition? The NiD-mediated reaction indicated that the NiD insertion is not the enantio-determining step. Is it possible that the isomerization to the desired enantiomer takes place at the Ni(I) species?

Currently we have no direct experimental evidence for the radical character of high-valent Ar-Ni(III)-alkyl species. However, previous workers have claimed the rapid homolysis process of high-valent Ar-Ni(III)-alkyl and subsequent enantioconvergent recombination under similar conditions (see refs 59-70), and this is our current working hypothesis. Currently there is no reported evidence about enantioconvergent isomerization of sec-alkyl Ni(I) species. Experiments towards the detection and characterization of Ni(III) species is still under investigation in our laboratory. Progress in this area will be reported in due course.

2. Could the authors give some explanation for the role of NaF while KF shows poor reactivity?

In accord with this comment, as well as a comment from reviewer 2 (*What is the role of Na⁺ ion in this reaction?*), we have now added a detailed conditions optimization in SI. As shown in Fig. S11 in Supplementary Information (page S34), when NiI₂ was used as nickel source, the best base is NaF (entry 12). However, when NiCl₂ was used as nickel source, the best base is KF (entry 10). We attribute the dramatic base effect to the following reason:

different nickel precatalysts require different base to efficiently form the same reactive $L^*Ni(I)F$ species to initiate the catalytic cycle.

3. Is there any chemoselectivity for C-I bonds with difference steric environment, such as 1,4-diiodo-2-methylbenzene?

As shown in page S31 in Supplementary Information, we have now carried out a competition experiment to compare the reactivity of aryl iodides with different steric environment. The least steric hindered aryl iodide has the best reactivity. We have now mentioned this information in the main text (at the substrate scope section).

4. In the Supporting Information, the ^{13}C NMR spectroscopies were not properly referenced. They are not inconsistent with the statement “are referenced $CDCl_3$ at 77.16 ppm”.

We have now referenced $CDCl_3$ at 77.16 ppm.

5. Page S136, the compound 1k was contaminated by impurities.

We have now re-purified the compound **1k** and re-characterized the corresponding 1H NMR and ^{13}C NMR spectra.

6. There are shoulder peaks in 3e, page S152. Please double check!

To rule out the possible shoulder peaks in the HPLC trace of **3e**, we have now re-purified the product and performed the HPLC trace under both original conditions (AD-H column) and different conditions (OD-H column). In both cases, the ee values are similar (98% ee). We have now provided both HPLC trace in SI.

In response to **reviewer 2** (quotes from reviewer are italicized):

Reviewer #2 (Remarks to the Author):

Zhu and co-workers have reported a method for the enantioselective synthesis of benzylic amine derivatives from vinyl enamines and aryl halides. The method is very simple and effective. Independently, a related work by the Nevado group just appeared in ACIE “doi.org/10.1002/anie.202011342”. However, reaction conditions used in this paper has much better reaction efficacy in terms of scope, functional group compatibility, yields, enantioselectivities as well as substrates (e.g. internal olefins are working well). The authors also show that the products can be obtained effectively independent of the E/Z isomers of the substrates. Given that operational simplicity, general applicability, valuable product formation, and surge of this emerging NiH catalysis area, I strongly recommend this work in Nature Communication after a minor revision.

Specific comments:

1. What about other N-Protected functional groups (e.g. NHCbz, NHBoc)?

As shown in Fig. S10 in Supplementary Information (page S33), when other N-protecting groups such as a Cbz and a Boc were used, low yields and poor regioselectivities were obtained.

2. “Evaluation of ligands showed that both the imidazoline skeleton (entry 3 vs entry 1) and the remote steric effects of the substituent on the imidazoline skeleton (entry 4 vs entry 3) have a dramatic influence on the enantioselectivity.” How does it affect the reactivity and enantioselectivity? Why bis-imidazoline ligand is more effective than bi-ox ligand? Some explanations are welcome.

Changing the oxazoline skeleton to imidazoline influences both the electron density and nucleophilicity of coordinating 3-nitrogen. For the detailed explanation of a long, branched, alkyl chain (4-heptyl) as the preferred substituent on the ligand, see Matthew S. Sigman and

Abigail G. Doyle's JACS paper, we have now added this paper as Ref 70.

3. What is the role of Na⁺ ion in this reaction?

As shown in Fig. S11 in Supplementary Information (page S34), when NiI₂ was used as nickel source, the best base is NaF (entry 12). However, when NiCl₂ was used as nickel source, the best base is KF (entry 10). We attribute the dramatic base effect to the different reactivities in the initiation step for different nickel precatalysts to form the same reactive L*Ni(I)F species to participate the catalytic cycle.

4. In SI, the yields of the synthesis of the substrates are missing.

We have now added the corresponding yields of the substrate synthesis.

5. How did the author establish the syn- and anti-configuration of the 4h-D?

Currently, it is difficult to identify the *syn*- and *anti*-configuration of the **4h-D**. We have now deleted the specific *syn*- and *anti*- assignment in Figure 5.

6. The authors have an intriguing observations. E- and Z isomers give similar EEs. Can the authors check if there are relevant isomerization processes?

Yes, during the reaction process, we could observe the isomerization of *E* olefinic substrate to *Z* isomer. However, the isomerization of *Z* olefinic substrate to *E* isomer is very slow when *Z* olefin was used (only a small amount of *E* isomer was observed). We have now added this information in the supplementary information (see page S29, section V in Supplementary Information).

REVIEWERS' COMMENTS

Reviewer #1 (Remarks to the Author):

I am satisfied the revision of this manuscript.
Accept.

Reviewer #2 (Remarks to the Author):

The authors have revised the paper according to the referees' comment. I am satisfied with the changes.